# Design of Ultra-Wideband Doherty Power Amplifier Using a Modified Combiner Integrated with Complex Combining Impedance

**DOI:** 10.3390/s23083882

**Published:** 2023-04-11

**Authors:** Jian Chen, Zhihui Liu, Tao Dong, Weimin Shi

**Affiliations:** 1Space Star Technology Co., Ltd., Beijing 100086, China; chenjian1@spacestar.com.cn (J.C.); liuzhihui1225@163.com (Z.L.); dongtaoandy@163.com (T.D.); 2School of Microelectronics and Communication Engineering, Chongqing University, Chongqing 400044, China

**Keywords:** complex combiner, Doherty power amplifier, modified combiner, ultra-wideband

## Abstract

To be compatible with future wireless communication systems, it is very necessary to extend the bandwidth of the Doherty power amplifier (DPA). In this paper, a modified combiner integrated with a complex combining impedance is adopted to enable an ultra-wideband DPA. Meanwhile, a comprehensive analysis is performed on the proposed method. It is illustrated that the proposed design methodology can provide power amplifier (PA) designers with more freedom in implementing ultra-wideband DPAs. As a concept of proof, a DPA working over 1.2–2.8 GHz (a relative bandwidth of 80%) is designed, fabricated and measured in this work. Experimental results showed that the fabricated DPA delivers a saturation output power of 43.2–44.7 dBm with a gain of 5.2–8.6 dB. Meantime, the fabricated DPA achieves a saturation drain efficiency (DE) of 44.3–70.4% and a 6 dB back-off DE of 38.7–57.6%.

## 1. Introduction

As the core component of the transmitter front-end, the power amplifier (PA) plays a vital role in wireless communication systems, such as mobile terminals, communication base stations, satellite communications and so on [1]. To fulfil ultra-speed data transmission, the wireless communication industry is undergoing great changes [2]. First, array communication becomes a necessary technical factor to improve system capacity and data rate [3]. This means power consumption further increases. Second, the modulation type of communication signals become more and more complex, such as 64 quadrature amplitude modulation (64-QAM) and 256-QAM modulation [4]. Third, the coverage band of the communication system becomes wider and wider [5]. The above three changes require PAs to have wider operating bandwidths, higher conversion efficiency and larger dynamic ranges. Active load modulation, such as that provided by the Outphasing power amplifier (OPA) and the Doherty power amplifier (DPA), is a promising technique to boost efficiency at the power back-off region [6,7,8]. Over the past few years, the DPA has been the most popular PA architecture used in wireless communication systems [8,9]. However, the bandwidth of the DPA needs to be further improved to support 5G/6G communications. Therefore, extending the bandwidth of the DPA is a challenging, but meaningful, pursuit [10,11,12,13].

In a DPA, the most important part is the impedance inverter, which is used to realize correct load modulation between the carrier and peaking PAs [9]. However, it has been demonstrated in many works that the impedance inverter significantly limits the bandwidth of the DPA [10,11]. After a series of studies, PA researchers found that the working bandwidth of the DPA can be extended by reducing the transformation ratio of the impedance inverter [12,13,14]. In [14], the authors designed a 0.55–1.1 GHz (an octave bandwidth) DPA by reducing the impedance ratio of the impedance inverter. Indeed, any passive elements between the carrier transistor and the impedance inverter limit the bandwidth of the DPA [15]. Therefore, the post-matching method was developed and investigated to broaden the bandwidth of the DPA [15,16]. In [16], the authors performed a comprehensive analysis of the post-matching DPA. It was demonstrated in [16] that a low order matching network is more suitable to implement broadband DPA. In [17], the authors implemented an asymmetrical DPA which achieved a drain efficiency (DE) of 51–55% at the 10 dB output back-off (OBO) power level over 1.6–2.2 GHz. Moreover, a post-matching DPA, based on a mutually-coupled harmonic matching network, was presented in [18] to improve DPA performance. Thus far, the post-matching technique has become the most popular method for designing broadband DPA [17,18,19,20,21,22,23,24,25,26].

On the other hand, the peaking PA also affects the bandwidth of a DPA, not only in the high power region but also in the low power region where the peaking PA is in the off state [19,20,21,22]. It was validated in [20] that the bandwidth of the DPA can be enhanced by means of careful tuning of the peaking impedance in the low power region. In [21], a broadband DPA working, over 1.7–2.8 GHz, was implemented, by inserting a quarter wavelength line into the output of the peaking path, to compensate for the carrier load at the low power region. This broadband DPA achieved a 6-dB back-off DE of 50–55% and a saturation DE of 57–71% in the frequency band of interest [21]. In [22], non-infinity peaking impedance was utilized to form a continuous-mode DPA, which could maintain high efficiency at a specified OBO power level over a wide bandwidth [23,24,25,26,27]. In [25], a broadband class-J DPA, operating from 2.7 GHz to 4.3 GHz, was designed, achieving a saturation DE of more than 48% and a 6 dB back-off DE of more than 40%.

In summary, the bandwidth of the DPA is determined by its combiner. Therefore, lots of novel combiners have been developed so as to design wideband DPAs [28,29,30,31,32,33,34,35,36]. In [28], an ultra-wideband DPA was realized, which was based on a pre-established combiner with a closed-form formulation. This ultra-wideband DPA maintains a 6 dB back-off DE of 35–58% in the frequency band of 1.05–2.55 GHz. In [31], the complex combining method was introduced into the Doherty combiner to extend the bandwidth. The complex combined DPA in [31] achieves a saturation power of 43.3-45.4 dBm, a saturation DE of 55.4–68% and a 6 dB back-off DE of 43.8–54.9% over 1.1–2.4 GHz. In [36], two λ/4 transmission lines, with negative characteristic impedance, were inserted into the Doherty combiner to extend the bandwidth of the DPA. An ultra-wideband DPA, covering 0.8–2.7 GHz, was implemented in [36], achieving a DE of 39.5–52% at the 6 dB OBO power level.

Unlike previous works, this paper presents a modified Doherty combiner integrated with complex combining impedance to design ultra-wideband DPAs. The proposed method can compensate the frequency dispersion of the Doherty combiner, leading to an extended bandwidth. The proposed design methodology provides power amplifier (PA) designers with more freedom in implementing ultra-wideband DPAs. To validate the proposed method, a DPA covering 1.2–2.8 GHz is implemented in this work.

The arrangement of this paper is as follows. The modified Doherty combiner with complex combining impedance is analyzed and presented in Section 2. A broadband DPA is designed and simulated in Section 3. Section 4 presents the experimental results of the fabricated DPA. Finally, the paper is concluded in Section 5.

## 2. Modified Doherty Combiner Integrated with Complex Combining Impedance

Traditionally, a DPA consists of two sub-amplifiers, a carrier PA D1 and a peaking PA D2, as shown in Figure 1a. In this figure, the carrier and peaking transistors are replaced by current sources $I_{1}$ and $I_{2}$, respectively. In practice, the utilized transistors always have intrinsic elements (parasitic and packaged elements). Therefore, the two sub-amplifiers of the DPA influence each other not only in the high power region, in which the peaking PA is switched on, but also in the low power region, in which the peaking PA is in the off state. Considering the above aspects, it was demonstrated in [27] that the most favorable combiner for broadband DPA is composed of a 90° transmission line (TL) and a 180° TL at the center frequency point, as shown in Figure 1a. Normally, the combining impedance of the DPA is $R_{L}$ = $Z_{0}$/2, where $Z_{0}$ is the optimal impedance of a class-B biased transistor. In practical design, the intrinsic elements of the utilized transistors should be absorbed into the $TL_{1}$ and $TL_{2}$. Though the DPA described in Figure 1a can maintain high efficiency over a wide bandwidth, a modification is necessary to further extend the bandwidth.

Based on the combiner shown in Figure 1a, a modified Doherty combiner is proposed in this work and illustrated in Figure 1b. The proposed Doherty combiner is composed of three impedance inverters, $TL_{1}$, $TL_{2}$ and $TL_{3}$. The characteristic impedance of these three impedance inverters are $Z_{1}$, $Z_{2}$ and $Z_{3}$, respectively. Furthermore, the combining impedance of the Doherty is no longer $Z_{0}$/2, but rather a complex impedance $Z_{L}$, as shown in Figure 1b. When designing a DPA, the characteristic impedance of the three impedance inverters ($Z_{1}$, $Z_{2}$ and $Z_{3}$) and the combining impedance ($Z_{L}$) can be tuned to extend the bandwidth. The proposed Doherty combiner is analyzed in the following sections.

### 2.1. General Equations

In Figure 1b, $Z_{C}$ and $Z_{P}$ are the load impedance of the carrier and peaking transistors. All the TLs in Figure 1b have an electrical length of 90° at the center frequency point $f_{0}$. As in previous works, $I_{1}$ is related to the input voltage, and can be expressed as:(1)$$\begin{array}{c} I_{1}=-g_{m}vin∠\left(-90^{{}^{\circ}}\cdot f\right). \end{array}$$
where $g_{m}$ is the trans-conductance of the transistor, vin is the normalized input voltage, and *f* is the normalized frequency (normalized to $f_{0}$). $g_{m}$ and vin are designated as $g_{m}$ = 1 and 0≤vin≤1 in this paper. Notice that, a 90° phase delay is introduced into $I_{1}$ to ensure in-phase power combining in the Doherty combiner.

For the peaking transistor, the current $I_{2}$ is related not only to the input voltage, but also to the switched-on time. To simplify the analysis, only symmetrical DPA was considered in this paper. Therefore, the peaking transistor is turned on at half the maximum input. Then, the current of the peaking transistor ($I_{2}$ in Figure 1b) can be expressed as [11]: (2)$$\begin{array}{c} I_{2}=\left\{\begin{array}{cc} 0,\; & vin\leq 1/2\\ -g_{m}\frac{vin-1/2}{1-1/2}, & vin>1/2 \end{array}\right. \end{array}$$

On the other hand, the ABCD-matrices of $TL_{1}$, $Z_{L}$, $TL_{2}$ and $TL_{3}$ are as follows [11]:(3)$$\begin{array}{c} \left[\begin{array}{cc} A_{1} & B_{1}\\ C_{1} & D_{1} \end{array}\right]=\left[\begin{array}{cc} \cos(\theta_{1}\cdot f) & jZ_{1}\sin\left(\theta_{1}\cdot f\right)\\ j\sin\left(\theta_{1}\cdot f\right)/Z_{1} & \cos(\theta_{1}\cdot f) \end{array}\right], \end{array}$$
(4)$$\begin{array}{c} \left[\begin{array}{cc} A_{L} & B_{L}\\ C_{L} & D_{L} \end{array}\right]=\left[\begin{array}{cc} 1 & 0\\ 1/Z_{L,i} & 1 \end{array}\right], \end{array}$$
(5)$$\begin{array}{c} \left[\begin{array}{cc} A_{2} & B_{2}\\ C_{2} & D_{2} \end{array}\right]=\left[\begin{array}{cc} \cos(\theta_{2}\cdot f) & jZ_{2}\sin\left(\theta_{2}\cdot f\right)\\ j\sin\left(\theta_{2}\cdot f\right)/Z_{2} & \cos(\theta_{2}\cdot f) \end{array}\right]. \end{array}$$
(6)$$\begin{array}{c} \left[\begin{array}{cc} A_{3} & B_{3}\\ C_{3} & D_{3} \end{array}\right]=\left[\begin{array}{cc} \cos(\theta_{3}\cdot f) & jZ_{3}\sin\left(\theta_{3}\cdot f\right)\\ j\sin\left(\theta_{3}\cdot f\right)/Z_{3} & \cos(\theta_{3}\cdot f) \end{array}\right]. \end{array}$$

Using Equations (Equation 1)–(Equation 6), the ABCD-matrix of the proposed Doherty combiner can be expressed as [27]:(7)$$\begin{array}{c} \left[\begin{array}{cc} A & B\\ C & D \end{array}\right]=\left[\begin{array}{cc} A_{1} & B_{1}\\ C_{1} & D_{1} \end{array}\right]\left[\begin{array}{cc} A_{L} & B_{L}\\ C_{L} & D_{L} \end{array}\right]\left[\begin{array}{cc} A_{3} & B_{3}\\ C_{3} & D_{3} \end{array}\right]\left[\begin{array}{cc} A_{2} & B_{2}\\ C_{2} & D_{2} \end{array}\right]. \end{array}$$

The relationships between the port voltages and currents can be expressed as:(8)$$\begin{array}{c} \left[\begin{array}{c} V_{1}\\ -I_{1} \end{array}\right]=\left[\begin{array}{cc} A & B\\ C & D \end{array}\right]\left[\begin{array}{c} V_{2}\\ I_{2} \end{array}\right]. \end{array}$$

Consequently, the drain voltages of the carrier and peaking transistors can be expressed as [27]:(9)$$\begin{array}{c} V_{1}=A\cdot V_{2}+B\cdot I_{2}, \end{array}$$
(10)$$\begin{array}{c} V_{2}=\frac{I_{1}+D\cdot I_{2}}{C}. \end{array}$$

After deriving the voltages and currents of the carrier and peaking transistors, the load impedance can be deduced as well as the output power and drain efficiency. By means of (Equation 1), (Equation 2), (Equation 9) and (Equation 10), the load impedance of the carrier and peaking transistors can be derived by using the following equations:(11)$$\begin{array}{c} Z_{C}=V_{1}/I_{1}, \end{array}$$
(12)$$\begin{array}{c} Z_{P}=V_{2}/I_{2}. \end{array}$$

Actually, the bandwidth of the DPA is mainly related to the load impedance of the carrier transistor at the low power region. In the following subsection, the load impedance of the carrier PA at the low power region is analyzed.

### 2.2. Analysis of the Modified Doherty Combiner

In the low power region, the peaking transistor is in the off state. Therefore, $I_{2}$ = 0 can be determined. Here, we define $Z_{CB}$ as the load impedance of the carrier transistor at the low power region. According to the above equations, $Z_{CB}$ is related to $Z_{1}$, $Z_{2}$, $Z_{3}$ and $Z_{L}$ as well as working frequency.

In the traditional DPA, $Z_{1}$ = $Z_{2}$ = $Z_{3}$ = $Z_{0}$ and $Z_{L}$ = 0.5$Z_{0}$. In this situation, the variation of $Z_{CB}$ versus normalized frequency is illustrated in Figure 2a. In this figure, the characteristic impedance of the Smith chart is $Z_{0}$. Figure 2a shows that $Z_{CB}$ changes significantly in the normalized frequency band of 0.6≤f≤1.4. The real part of $Z_{CB}$ is much less than 2$Z_{0}$ at the two sidebands. This leads to degradation in the back-off efficiency. Therefore, it is almost impossible for the traditional DPA to maintain high efficiency over the normalized frequency band of 0.6≤f≤1.4.

Fortunately, a wider bandwidth is obtained by the DPA if the parameters of the combiner are carefully selected. When $Z_{1}$ = 1.13$Z_{0}$, $Z_{2}$ = 1.4$Z_{0}$, $Z_{3}$ = $Z_{0}$, and $Z_{L}$ = 0.5$Z_{0}$, the calculated $Z_{CB}$ over 0.6≤f≤1.4 is depicted in Figure 2b (the dashed blue line). In this situation, the real part of $Z_{CB}$ is improved at the two sidebands compared to the traditional DPA. In this way, the back-off DE of the DPA is enhanced.

Furthermore, when $Z_{1}$ = 1.13$Z_{0}$, $Z_{2}$ = 1.4$Z_{0}$, $Z_{3}$ = $Z_{0}$, and $Z_{L}$ = 0.6$Z_{0}$, the calculated $Z_{CB}$ over 0.6≤f≤1.4 is also shown in Figure 2b (the red line). In this case, the real part of $Z_{CB}$ is close to 2$Z_{0}$ over the entire frequency band of interest (0.6≤f≤1.4).

Notice that, in the above situations, the combining impedance $Z_{L}$ is set to a pure real number. Actually, the bandwidth of the DPA can be further enhanced by setting $Z_{L}$ to be complex over the frequency band of interest [31]. Figure 2c shows the load impedance of the carrier transistor at the low power region when the normalized frequency is *f* = 0.6. In Figure 2c, $Z_{1}$ = 1.13$Z_{0}$, $Z_{2}$ = 1.4$Z_{0}$, and $Z_{3}$ = $Z_{0}$. It can be observed in Figure 2c that $Z_{CB}$ moves down to the real axis as the imaginary part of $Z_{L}$ becomes smaller.

In summary, the load impedance trajectory of the carrier transistor at the low power region can be manipulated by elaborately selecting the parameters of the proposed Doherty combiner. Therefore, broadband DPA can be implemented by carefully designing the four parameters ($Z_{1}$, $Z_{2}$, $Z_{3}$ and $Z_{L}$). Moreover, these four parameters of the proposed combiner supply more freedom to DPA designers.

## 3. Design and Simulation of a Broadband DPA

Based on the above analysis, the design and simulation of a broadband DPA is presented in this section. The center frequency $f_{0}$ of the designed DPA was set to 2 GHz. The frequency band of interest was 1.2–2.8 GHz (0.6≤f≤1.4). The transistors utilized in this design are CGH40010Fs from Wolfspeed. The optimal impedance of this kind of transistor is set to $Z_{0}$ = 30 Ω [27]. All the simulations using this design were based on the Rogers 4350B substrate, with a thickness of 20 mil. To obtain apparent Doherty behavior and high performance, the carrier and peaking transistors should be biased in deep class-AB and class-C conditions, respectively. The gate voltages of the carrier and peaking transistors are −3 V and −5.8 V, respectively. Meanwhile, the drain bias voltages of the transistors are both set to 28 V.

### 3.1. Design

To cover the frequency band of interest, the parameters of the proposed Doherty combiner were set to $Z_{1}$ = 34 Ω, $Z_{2}$ = 42 Ω, and $Z_{3}$ = 30 Ω. These design parameters were derived by performing a simulation in advanced design system (ADS) software. Furthermore, the combining impedance $Z_{L}$ changes versus working frequency in this design. The whole schematic of the designed DPA is shown in Figure 3. The design procedure of the DPA was as follows:

Firstly, the impedance inverter $TL_{1}$, with a characteristic impedance of 34 Ω, is replaced by a matching network which takes the intrinsic elements of the carrier transistor into consideration. In other words, the impedance inverter, after the carrier transistor, is composed of the intrinsic elements and a matching network. Figure 4a depicts the schematic of the composite impedance inverter $TL_{1}$. The simulation results of the composite $TL_{1}$ with two 34 Ω terminations are shown in Figure 4b. This figure indicates the phase delay of the composite $TL_{1}$ at 2.0 GHz was roughly −90°. Meanwhile, the simulated S11 of the composite $TL_{1}$ was less than −20 dB over the entire frequency band of interest.

Secondly, the intrinsic elements of the peaking transistor also produce effects on the impedance inverter $TL_{2}$. Therefore, the length of $TL_{2}$ was reduced to compensate for the phase delay of the intrinsic elements, as depicted in Figure 3. Moreover, a 42 Ω transmission line, with a phase delay of 90° ($TL_{3}$) at 2 GHz, was inserted between $TL_{2}$ and the combining point. Theoretically, the width and length of $TL_{3}$ were 1.4 mm and 21 mm, respectively. Nevertheless, considering the effect of the *T* junction, the length of the impedance inverter $TL_{3}$ was reduced to 18 mm, as shown in Figure 3.

Thirdly, a post-matching network (PMN) is constructed to transform the 50 Ω standard load to the complex combining impedance, as shown in Figure 3. Notice that the complex combining impedance is designed such that a wider bandwidth can be obtained by the DPA. To obtain the optimal combining impedance, optimization was performed in the Advanced Design System (ADS) when designing the DPA. The schematic of the designed PMN is shown in Figure 5. This figure also shows the simulated combining impedance of the Doherty combiner versus working frequency. The real part of the combining impedance changes from 16.8 Ω to 23.9 Ω in the frequency band of interest. The imaginary part of the combining impedance varies from −2.5 Ω to 3.4 Ω. To validate the designed PMN, the whole combiner was simulated when the combining point was connected to the traditional combining impedance 15 Ω and the designed PMN, respectively. The simulation schematic is illustrated in Figure 6a. In this figure, the intrinsic elements of the utilized transistors are also considered. The simulated $Z_{CB}$ over 1.2–2.8 GHz is also shown in Figure 6b. This figure indicates that the complex combining impedance further extended the bandwidth of the DPA.

Finally, to enable better performance, two input matching networks (IMNs) were designed for the carrier and peaking PAs, as shown in Figure 3. Optimization was performed when designing the IMNs. Moreover, a two-stage Wilkinson power divider is utilized to split input power equally to the two sub-amplifiers. To ensure in-phase power combining, a phase offset line is inserted before the carrier IMN, as shown in Figure 3. The design process of the broadband DPA has now been fully described.

### 3.2. Simulation

Using the simulation schematic shown in Figure 3, the designed DPA was simulated and evaluated. The simulated load modulation trajectories of the carrier transistors at some frequencies are shown in Figure 7. In this figure, the load modulation trajectories were obtained at the transistor internal plane via de-embedding technique. At the low power region, the real part of $Z_{C}$ was close to 2$R_{0}$, as shown in Figure 7. As the input power increased, $Z_{C}$ modulated to $Z_{0}$.

After simulating the load modulation trajectories of the carrier transistor, the DEs and gains of the designed DPA were simulated and are illustrated in Figure 8a,b, respectively. These two figures show that obvious Doherty behavior was obtained by the designed DPA over the whole frequency band of interest. Figure 8a shows that the designed DPA achieved a saturation output power of more than 43 dBm with a saturation DE of more than 50%. Meanwhile, the designed DPA maintained a DE of more than 40% at the 6 dB back-off power level. Moreover, Figure 8b shows that the simulated gain of the designed DPA was larger than 9 dB at the saturation power level.

Figure 9 shows the simulated output power and DE of the designed DPA versus frequency. The red line with dots in Figure 9 indicates the designed DPA delivered an output power of 43.0–44.9 dBm at the saturation power level. The green line in Figure 9 shows that the DE of the designed DPA at the saturation power level was 51.3–68.9% over 1.2–2.8 GHz. The dashed blue line in Figure 9 shows that the designed DPA maintained a DE of 41–55.3% in the frequency band of interest.

## 4. Experimental Results

To validate the proposed method, the designed DPA was fabricated and tested. A photograph of the fabricated broadband DPA is shown in Figure 10. When testing the fabricated DPA, the gate voltages of the carrier and peaking transistors were −2.8 V (quiescent current was 30 mA), and −6 V, respectively. The drain voltages of both the carrier and the peaking transistors were 28 V.

Figure 11a shows the measured output power of the fabricated DPA versus input power over 1.2–2.8 GHz with a step of 0.2 GHz. The fabricated DPA delivered a maximum output power of more than 43 dBm in the frequency band of interest. The measured gains of the fabricated DPA versus output power are shown in Figure 11b. This figure indicates that the gain of the fabricated DPA was 5.2–8.6 dB over 1.2–2.8 GHz.

The measured DEs and power added efficiencies (PAEs) of the fabricated DPA with respect to output power over the frequency band of interest, with a step of 0.2 GHz, are illustrated in Figure 12a,b, respectively. Figure 12a shows the fabricated DPA achieved a saturation DE of more than 45%. Meanwhile, obvious efficiency enhancement was measured for the fabricated DPA. Figure 12b illustrates the measured PAE of the DPA was greater than 35.8% at the saturation power level over the whole frequency band of interest. The highest PAE (62.8%) was obtained at 2.6 GHz, while the lowest PAE (35.8%) was measured at 1.8 GHz.

To clearly observe the performances of the fabricated DPA, the measured output power and DE versus frequency are depicted in Figure 13. The red line in Figure 13 shows that the fabricated DPA delivered a saturation output power of 43.2–44.7 dBm over 1.2–2.8 GHz. The green line in Figure 13 indicates that the fabricated DPA achieved a saturation DE of 44.3–70.4% in the frequency band of interest. The blue line in Figure 13 shows the measured 6 dB back-off DE of the fabricated DPA was 38.7–57.6% over 1.2–2.8 GHz. Finally, the experimental results of the fabricated DPA are listed in Table 1. Meantime, the experimental results of some previous state-of-the-art DPAs are also given in Table 1 for comparison. According to the comparison table, a comparable performance was obtained by the proposed DPA over a relative bandwidth (RBW) of 80%.

Notice the differences between the simulation and experimental results, as shown in Figure 9 and Figure 13. These differences may have been caused by a process error in the printed circuit board (PCB) and the soldering of transistors, capacitors and other elements.

## 5. Conclusions

This paper proposes a modified Doherty combiner to design broadband DPAs. In the modified Doherty combiner, complex combining impedance is utilized to provide the PA designer with more design freedom. The proposed design methodology is suitable for implementing ultra-wideband DPAs. A DPA of 1.2–2.8 GHz (over an octave bandwidth) was designed and fabricated in this work to validate the proposed combiner. The experimental results showed that the fabricated DPA achieves a saturation output power of 43.2–44.7 dBm, a saturation DE of 44.3–70.4% and a 6 dB back-off DE of 38.7–57.6% over the entire frequency band of interest.

## Figures and Tables

**Figure 1 sensors-23-03882-f001:**
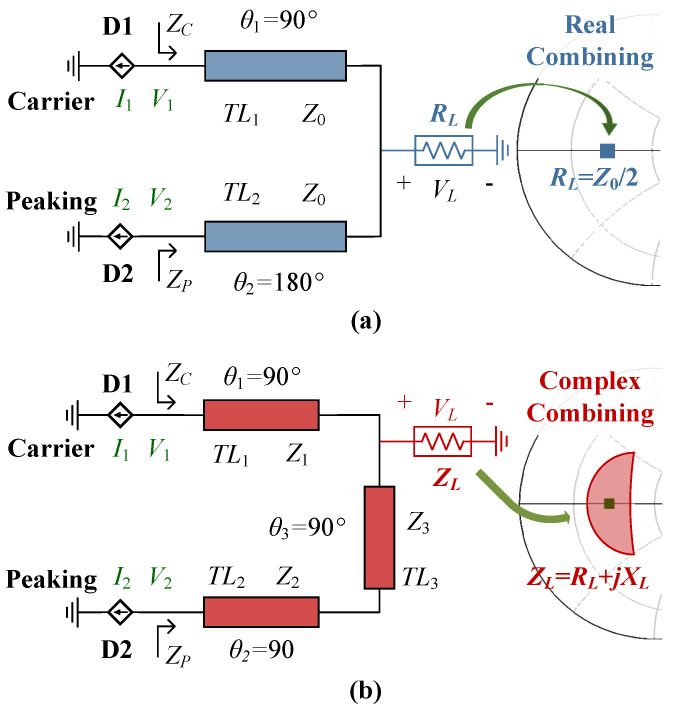
Simplified block diagrams of (**a**) the traditional DPA and (**b**) the proposed DPA.

**Figure 2 sensors-23-03882-f002:**
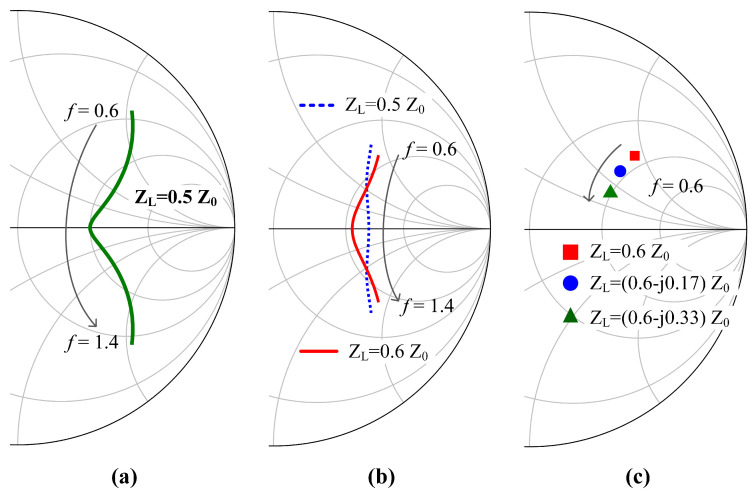
Calculated $Z_{CB}$ of (**a**) the traditional DPA and (**b**) the proposed DPA with modified combiner in the normalized frequency band of 0.6–1.4. (**c**) Calculated $Z_{CB}$ of the proposed DPA with complex combining impedance when *f* = 0.6.

**Figure 3 sensors-23-03882-f003:**
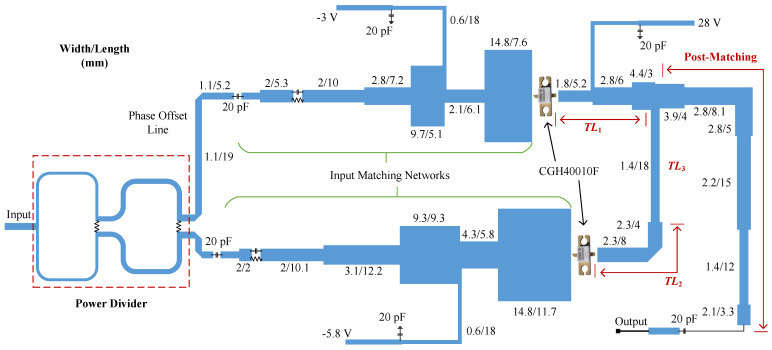
Schematic of the designed DPA together with the dimensions of the utilized passive elements.

**Figure 4 sensors-23-03882-f004:**
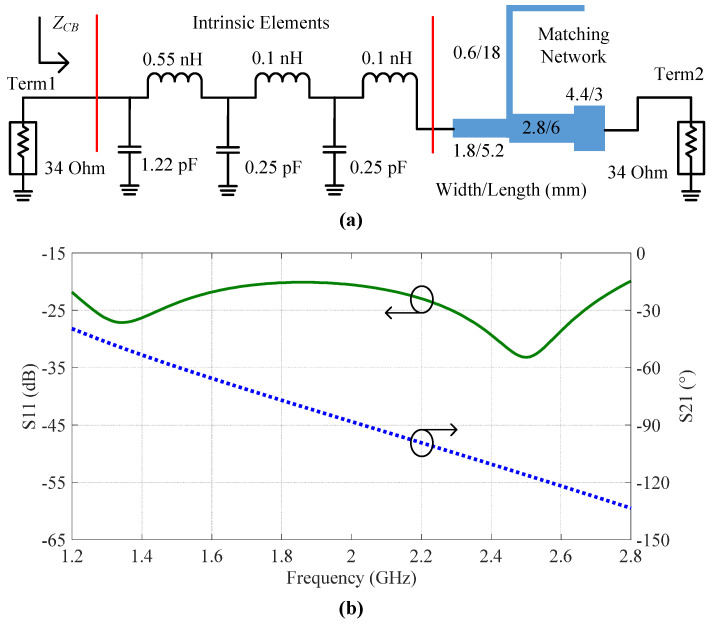
(**a**) Schematic and (**b**) simulation results of the impedance inverter *TL*_1_, which is composed of the intrinsic elements of the carrier transistor and a matching network.

**Figure 5 sensors-23-03882-f005:**
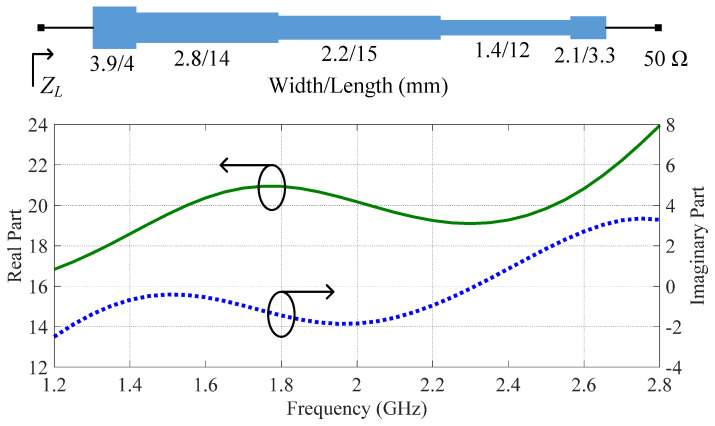
Schematic of the designed PMN and its simulation results.

**Figure 6 sensors-23-03882-f006:**
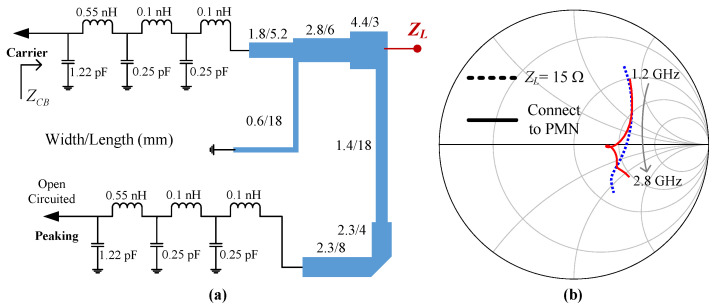
(**a**) Simulation schematic and (**b**) results of the proposed combiner using complex combining impedance.

**Figure 7 sensors-23-03882-f007:**
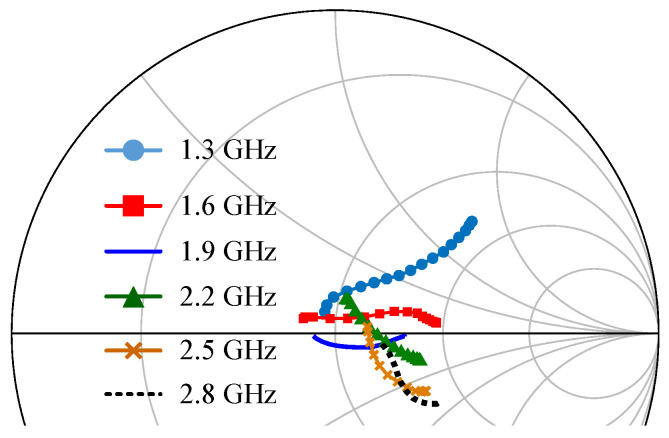
Simulated load modulation trajectories of the carrier transistor at the internal plane.

**Figure 8 sensors-23-03882-f008:**
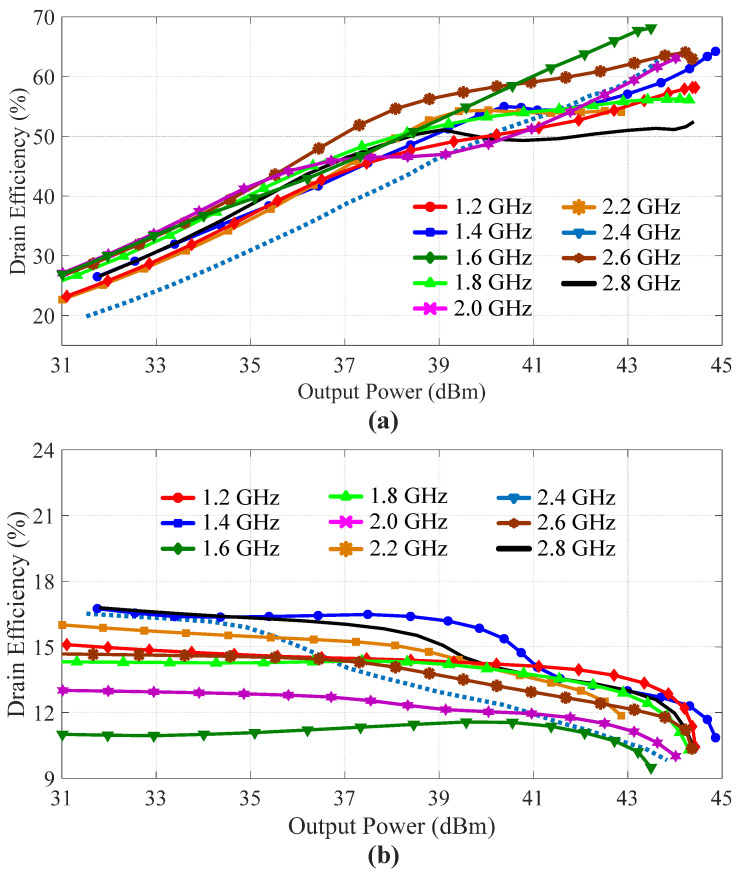
Simulated (**a**) drain efficiencies and (**b**) gains of the designed DPA versus output power at different frequencies.

**Figure 9 sensors-23-03882-f009:**
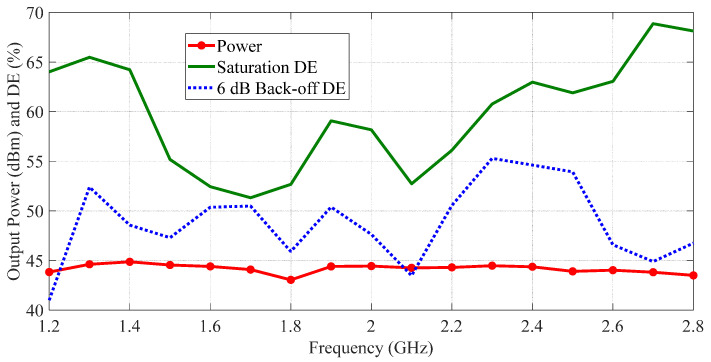
Simulated output power and drain efficiency of the designed DPA versus frequency.

**Figure 10 sensors-23-03882-f010:**
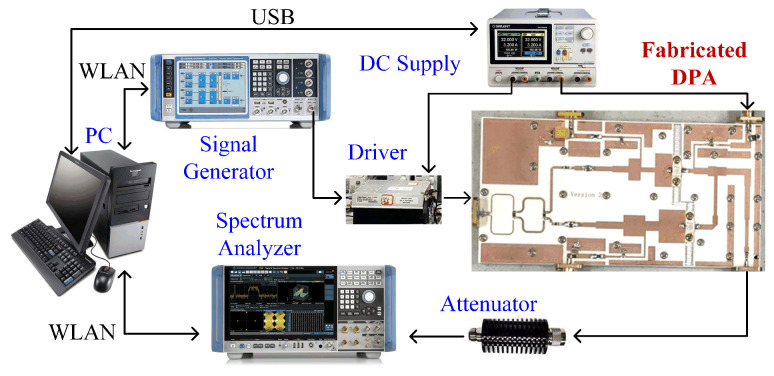
Photograph of the fabricated DPA and its experimental environment.

**Figure 11 sensors-23-03882-f011:**
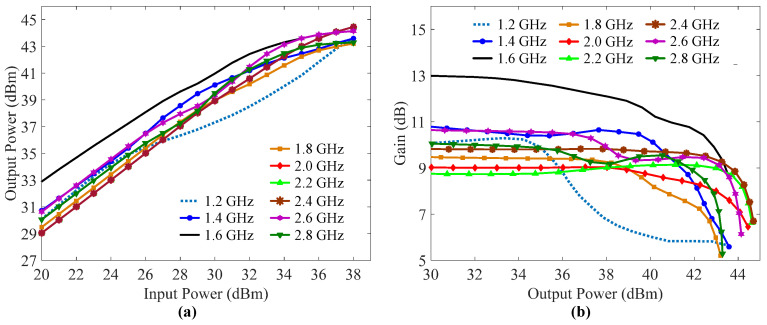
Measured (**a**) output power and (**b**) gains of the fabricated DPA at different frequencies.

**Figure 12 sensors-23-03882-f012:**
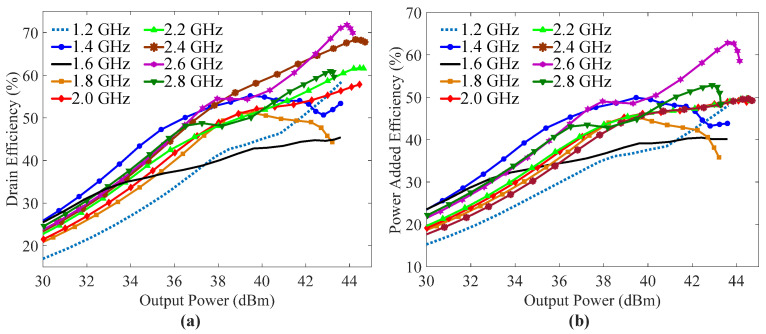
Measured (**a**) drain efficiencies and (**b**) gains of the fabricated DPA versus output power at different frequencies.

**Figure 13 sensors-23-03882-f013:**
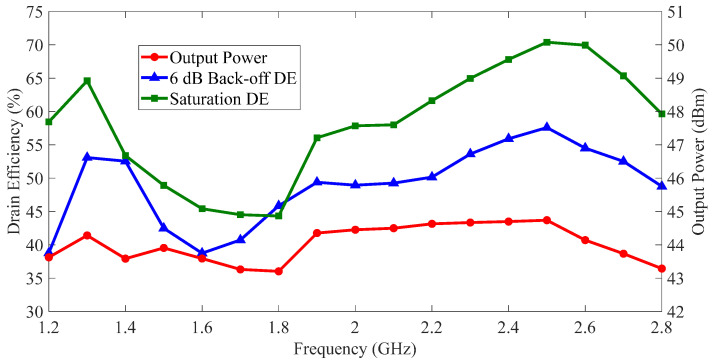
Measured output power and drain efficiency of the fabricated DPA versus frequency.

**Table 1 sensors-23-03882-t001:** Experimental results of the fabricated DPA and some previous state-of-the-art DPAs.

Ref. (Year)	Frequency (GHz)	RBW	Power (dBm)	Gain (dB)	DE@Sat (%)	DE@-6dB (%)
[14] (2017)	0.55–1.1	66.7%	42–43.5	N/A	56–72	40–52
[19] (2016)	1.5–2.5	50%	42–44.5	8–11	55–75	42–53
[21] (2016)	1.7–2.8	49%	44–44.5	11–12	57–71	50–55
[23] (2016)	1.65–2.75	50%	44–46	7–8	60–75	50–60
[24] (2019)	1.25–2.3	59.2%	41.4–44.6	N/A	56–75.4	45–56.5
[27] (2021)	1.5–2.55	51.8%	42.6–44.4	7.2–11.6	50.7–69.7	43.3–57
[28] (2014)	1.05–2.55	83%	40–42	>7	45–83	35–58
[30] (2018)	1.5–3.8	87%	42.3–43.4	10–13.8	42–63	33–55
[31] (2019)	1.1–2.4	74%	43.3–45.4	9.5–11.1	55.4–68	43.8–54.9
[33] (2020)	2.8–3.55	23.6%	43–45	8.3–9.1	66–78	50–60.6
[36] (2023)	0.8–2.7	108.6%	41.8–44	7.1–11.1	47.6–84.4	39.5–52
This Work	1.2–2.8	80%	43.2–44.7	5.2–8.6	44.3–70.4	38.7–57.6

DE@Sat: DE at saturation power level. DE@-6dB: DE at 6 dB OBO power level.

## Data Availability

The data presented in this study are included within the article.

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
