# Peer review of "Design of Ultra-Wideband Doherty Power Amplifier Using a Modified Combiner Integrated with Complex Combining Impedance"

_sensors, 2023, doi:10.3390/s23083882_

Round 1

Reviewer 1 Report

In this paper, a modified combiner  integrated with a complex combining impedance is adopted to enable an ultra-wideband DPA. Good work, but please carefully improve the manuscript by following comments.

1. Can the author provide a comparison of bandwidth characteristics between the proposed structure and traditional DPA?

2.  If the voltage current relationship between the main circuit and the peak circuit is given in Eq. (7), how can I calculate the matching impedance at the current source plane, and how can I calculate the efficiency curves?

3. From the simulation curve, the efficiency at the back-off point is very low (40%) for some frequency points, but from theoretical analysis, the efficiency can be very high. Please explain.

Reviewer 2 Report

The paper presents a slightly different take on the design of the  output matching for a Doherty PA structure. The work fails to demonstrate clearly what the added value of the approach is. The gain is quite low and clear comparison with an “ordinarily designed” structure is missing. Non-scientific terms are used here and there, such as ‘optimum’ with out stressing in what sense. I miss more clear description of the measurement setup and procedure. Several figures have template text for caption indicating that this is not publication ready work. Language needs minor upgrading. 

In conclusion .. incremental work at most and the paper fails to describe - to a satisfactory degree - what is novel and why this is a better approach.

Reviewer 3 Report

The authors have designed, fabricated and measured an ultra-wideband Doherty power amplifier using a modified combiner. The manuscript is well written and technical aspects are considered in the manuscript. The manuscript can be accepted after some modifications, which are listed as follows:

-        Kindly explain more about the DC voltage of peaking amplifier.

-        The obtained gain and efficiency parameters are rather low according to the applied transistor GaN CGH40010F, which is usually considered as high efficiency and gain transistor. Kindly explain more about this issue.

-        More related work should be cited in the introduction. As a suggestion: “A planner Doherty power amplifier with harmonic suppression with open and short ended stubs. Frequenz. 2022 Apr 26;76(3-4):121-30.”

-        Indicate the obtained P1db of the amplifier.

-        Po/pin parameter should be reported.

-        PAE is not mentioned in the manuscript.

-        The used Equations are rather general. Also, the used equations should be cited.              

-        Kindly mention the unit of dimensions in Figure 4
